# Optimisation of a convolutional neural network to segment the first trimester placenta from 3D ultrasound scans.

**Pádraig Looney**[1]    **Gordon N. Stevenson**[2]    **Kypros H. Nicolaides**[3]    **Walter Plasencia**[4]

**Malid Molloholli**[5,6]                          **Stavros Natsis**[5]

**Sally Collins**[1,5]

[1]Nuffield Department of Women's & Reproductive Health, University of Oxford, Level 3, Women's Centre, John Radcliffe Hospital, Oxford OX3 9DU.
[2] School of Women's and Children's Health, University of New South Wales, Randwick, NSW, Australia.
[3] Harris Birthright Research Centre of Fetal Medicine, King's College Hospital, UK.
[4] Fetal Medicine Unit. Hospiten Group. Tenerife. Canary Islands. Spain
[5] Fetal Medicine Unit, The Women's Centre, John Radcliffe Hospital Oxford
[6] Department of Obstetrics and Gynaecology, Wexham Park Hospital, Slough

## Abstract

Screening for increased risk of pregnancy complications could be possible with fully automated placental segmentation in 3D ultrasound (3D-US). Fully convolutional neural networks (fCNN) have previously obtained good segmentation performance of the first trimester placenta and appears to predict fetal growth restriction better than manual segmentation methods. The goal of this study is to adjust fCNN architecture parameters to investigate their impact on performance and ultimately to produce a more accurate segmentation. 2,393 first trimester 3D-US volumes with 'ground-truth' segmentation obtained using a semi-automated technique were used. An open source package (OxNNet) was used to train end-to-end six fully convolutional neural networks with different loss functions, addition of batch normalisation and with different numbers of features. A small increase in performance of placental segmentation in terms of Dice similarity coefficient (DSC) (0.835 vs 0.825) was observed. Doubling the feature map gave a minor improvement in DSC (0.01). Use of batch normalisation increased the speed of training as expected. The Dice-based loss gave poorer performance in general. Convolution with no padding produced better segmentation than using padding. The subjective case quality assessment score was shown to correlate with the DSC (r = -0.28 (p < 0.05)). A faster, less-memory intensive fCNN architecture can provide a similar segmentation performance moving the use of this tool for clinical screening a step closer.

## 1 Introduction

State of the art organ segmentation has been achieved using deep learning in a range of medical imaging modalities [Litjens et al., 2017]. 3D ultrasound however, has not been heavily studied. The first efforts to segment the placenta in early stage pregnancy were reported by Looney et al. [2017] using Deepmedic [Kamnitsas et al., 2017] on 300 cases and obtained a Dice similarity coefficient

1st Conference on Medical Imaging with Deep Learning (MIDL 2018), Amsterdam, The Netherlands.

(DSC) of 0.73. Yang et al. [2017] used 104 cases and obtained a DSC of 0.64 for the placenta. A recent study by Anonymous [2018] using 1197 cases and a modified U-Net [Ronneberger et al., 2015] architecture has obtained a DSC of 0.84.

Several clinical studies have demonstrated that a low placental volume (PlVol) in the first trimester can predict adverse outcomes later in the pregnancy including small for gestational age (SGA) babies [Collins et al., 2013] and pre-eclampsia[Hafner et al., 2006]. Other studies have demonstrated PlVol to be independent of other biomarkers for SGA such as pregnancy associated plasma protein A (PAPP-A) [Law et al., 2009, Collins et al., 2013] and nuchal translucency [Collins et al., 2013], which led a recent systematic review to conclude that it could be successfully integrated into a future multivariable screening method for SGA [Farina, 2016]. This would be similar to the 'combined test' currently used to screen for fetal aneuploidy. As PlVol could be measured at the same gestation as the routinely offered 'combined test', no extra ultrasound scans would be required making it economically appealing to healthcare providers worldwide.

It was shown by Anonymous [2018] that the automatic placental volume measured using a trained neural network is as effective as a semi-automated method of calculating placental volume in detecting SGA. The automatic measurement of placental volume also outperforms the proprietary manual VOCAL$^{TM}$(Virtual Organ Computer-aided AnaLysis; GE Healthcare, Milwaukee, WI) system. Improving the performance of the neural network will provide more accurate placental segmentations and so should improve the predictive value of PlVol.

There are many hyperparameters that can be modified to improve performance [Bengio, 2012] such as the number of feature maps. The choice of loss function has been shown to affect the accuracy of a CNN. Milletari et al. [2016] used a Dice based loss function to segment the prostate and showed that this loss function gave better performance than traditional cross-entropy loss (DSC of 0.869 and 0.739 for the Dice and cross-entropy loss respectively). Another possible source of improvement is the addition of batch normalisation which has been found to speed up training and have a regularising effect Ioffe and Szegedy [2015].

The output from a convolutional layer with no padding is reduced in size with respect to the input depending on the kernel size and striding. Ronneberger et al. [2015] used no padding in the 2D U-Net while the work by Milletari et al. [2016] in extending U-Net to 3D used padding. Padding reduced the computation cost since less segments are needed to perform full inference. However, it introduces issues with translation invariance since the confidence of voxel classification can depend on the proximity of a voxel from the edge in a segment.

In this work, we investigate the role of loss function, batch normalisation, padding and the number of feature maps on the performance of the neural network at segmenting the placenta in 3D ultrasound. Clinical assessment of the image quality is also compared to the prediction of the best performing of the models.

## 2   Methods

Plasencia et al. [2011] originially use the 3D-US data to assess the predictive value of PlVol, measured using the commercial VOCAL$^{TM}$ tool. All participants provided written consent and the study had full local ethical approval (ID:02-03-033). At 11+ 0 to 13 + 6 weeks' gestation when the women presented for their combined screening for aneuploidies at the Fetal Medicine Centre, London, UK [Kagan et al., 2008, Snijders et al., 1998] a 3D-US volume containing the placenta was recorded for 3,104 singleton pregnancies. The 3D-US volume was acquired by trans-abdominal sonography using a GE Voluson 730 Expert system (GE Medical Systems, Milwaukee, Wisc., USA) with a 3D RAB4/8L transducer Wegrzyn et al. [2005]. 336 of the original 3,104 3D-US volumes were discarded as they had been saved using wavelet-compression, which results in significant loss of the underlying raw data thereby preventing further analysis. Another 375 cases had been collected with the B-Mode gain set exceptionally high and were excluded. This gain setting is used in clinical practice as it makes the nuchal translucency more obvious but is inappropriate for imaging the placenta as it removes the subtle variation in the echogenicity of tissues resulting in a 'stark', black and white image appearance.

The Random Walker (RW) algorithm, which has been described previously [Stevenson et al., 2015], was used to annotate the remaining 2,393 3D-US volumes. 3D B-mode data was converted from the toroidal geometry GE Voluson Kretzfile format into a 3D Cartesian volume with isotropic 0.6 mm

spacing as described by [Looney et al., 2017]. The segmentation was initialised or 'seeded' by an operator (SN). These 'seedings' were then examined for accuracy by a second, independent, operator (MM) and 're-seeded' where mistakes were evident. Cases where there was uncertainty regarding the boundaries of the placenta were examined by a third operator (SC). The 'seedings' were then used to calculate the placenta segmentation using the RW method. Finally the 'ground-truth' dataset was quality controlled by visually inspecting the segmentation of all the cases seen to be outliers in the distribution of PlVol values. This was performed by three operators (SC, PL and GS), if an error was seen in the segmentation the seeding was checked and the image re-seeded and re-segmented where appropriate. The resulting 2,393 quality controlled 'ground-truth' segmentations were then used to train, validate and test the models.

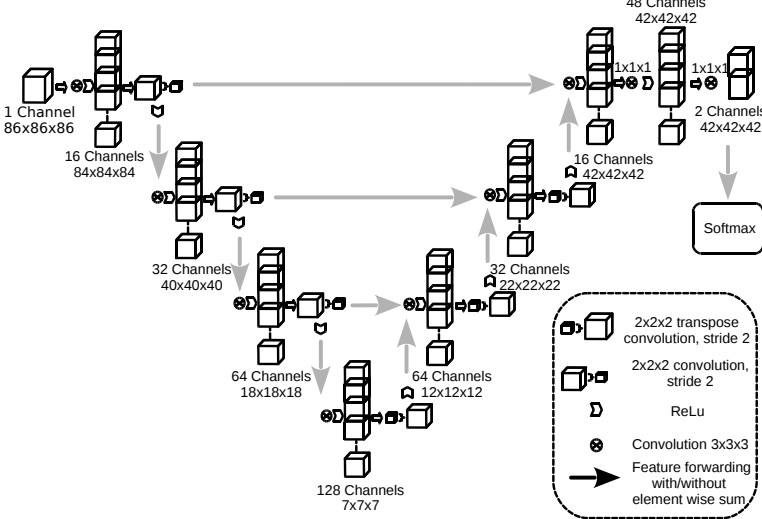

Figure 1: Architecture of model presented by Anonymous [2018]. This architecture was varied to improve the automatic segmentation of the placenta in 3D ultrasound.

The architecture used is based on 3D extension of the U-net architecture Ronneberger et al. [2015]. Using the architecture shown in figure 1 a DSC of 0.84 was obtained by Anonymous [2018]. The loss function used were the cross entropy loss and the modified Dice loss as described by Milletari et al. [2016]. The Dice loss used here was

$$loss = 1 - \Sigma_{classes} \frac{2 \times \Sigma_i^{voxels} P_i L_i}{\Sigma_i^{voxels} \left(P_i^2 + L_i^2\right) + \epsilon}, \tag{1}$$

where $\epsilon$ is 0.0001, $P_i$ is the softmax prediction for voxel $i$ and $L_i$ is the label for voxel $i$. For some of the models batch normalisation was added to the architecture shown in Figure 1 at all of the convolutional layers with stride one and kernel width of three. This was performed for both loss functions. The number of features of each layer of the architecture shown in Figure 1 were doubled and halved.

1,096, 100, and 1,197 cases were selected randomly from the 2,393 cases for training, validation and testing. The neural networks were trained for ten epochs. The models were realized using TensorFlow (version 1.5) Abadi et al. [2016] and OxNNet [Looney, 2013]. Cubic patches of 863 voxels were extracted from the full volumes and used as input to the CNN. The batch size was 20, 30 and 40 for the CNN with 32, 16 and 8 features in the first convolutional layer respectively. The parameters of the Adam optimizer learning rate, $\beta_1$, $\beta_2$ and $\epsilon$ were set as 0.001, 0.9, 0.999 and $1 \times 10^{-8}$ respectively. To reduce overfitting, dropout with probability 0.5 was applied to the final layer. Validation of a single batch of patches was performed every 50 steps and full validation on 100 cases was carried out at the end of the each epoch. After training the model was tested on 1,197 cases. The predicted segmentations were post-processed to remove disconnected parts of the segmentation less than $40\%$ of the volume of the largest region. The segmentation was binary dilated and eroded using a 3D kernel of radius three voxels and a hole filling filter applied.

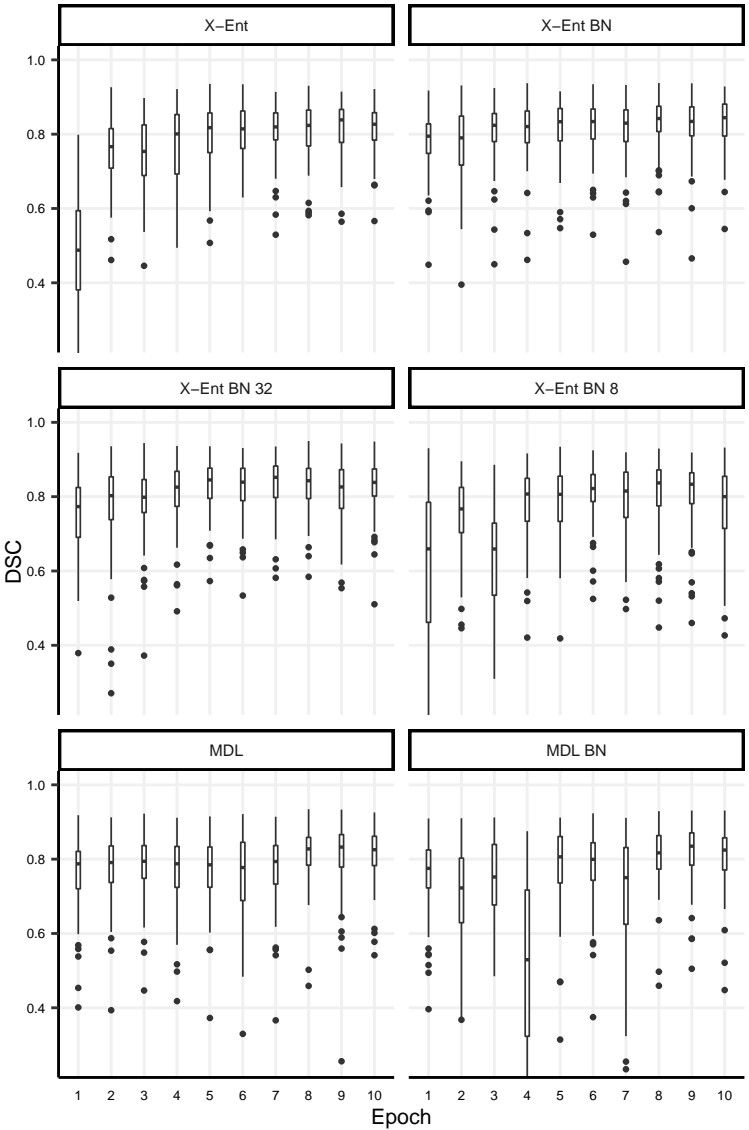

Figure 2: Validation of the 100 cases for each epoch during training for the different architectures.

Batch normalisation (BN) was added to the model previously reported by Anonymous [2018] (X-Ent) to produce the model X-Ent BN. The number of features of X-Ent BN was doubled (X-Ent BN 32) and halved (X-Ent BN 8). X-Ent was retrained using the modified Dice loss (DL) to give a different model MDL. This model was also trained using batch normalisation to produce the model MDL BN. In these models no padding was used in the convolutional layers, the cubic input was $86^3$ voxels in size which gave a cubic output of $42^3$ voxels in size. Finally two models identical to X-Ent BN but with padding in the convolutional layers were trained and tested using input and output of $48^3$ voxels in size (Patch 48 X-Ent BN) and $72^3$ voxels in size (Patch 72 X-Ent BN).

The images were assessed by an experienced operator (SN) and a scoring system was used to assess quality. Cases where over $20\%$ of the placenta volume was missing were scored two and cases where less than $20\%$ of the placenta volume was missing were scored one. The shape of the placenta can effect the difficulty of segmentation. Easy shaped placentas (e.g. isolated, globular shape and pancake shape placentas) were scored one while difficult shape placentas (e.g. c-shaped, unclear shape and placenta over fibroid) were scored two. Placentas whose appearance or composition was considered homogenous scored one, placentas whose composition was heterogeneous were scored two and

placentas with cystic areas or placenta lakes were scored three. The quality of the image was rated from one (contrast excellent, no pixelation, sharp image) to five (blurred image, very pixelated and difficult to assess boundaries). Images were scored two if the placenta was obscured or one otherwise. Segmentation of a placenta without any fetal parts touching or close to the placenta were much easier to segment. Those with no fetal parts adjacent or touching the placenta scored one point, placentas with a fetus body close to the placenta scored two points and with placentas where the whole of the fetus was close to the placenta were scored three points. The overall image quality score was then calculated by adding these six scores together.

## 3 Results

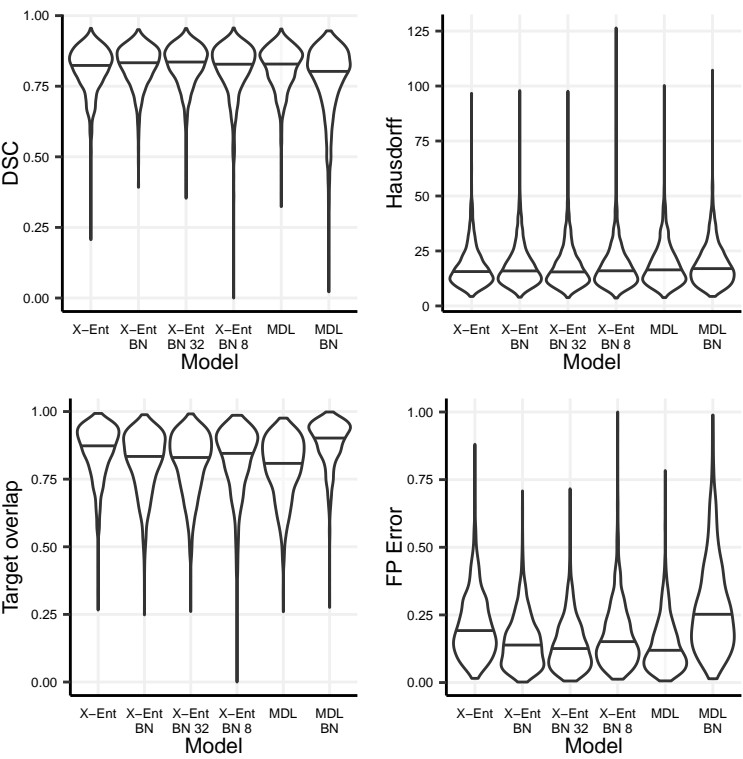

Figure 3: Distributions of the metrics measured for the different architectures on 1197 test cases.

Box plots of the DSC of the predicted segmentations for the 100 validation cases are shown in Figure 2 for each model and epoch. The best performing model for each of the architectures in Figure 2 was evaluated on the test cases.

Table 1: Model Metrics

| Model | DSC | Hausdorff (mm) | Target Overlap | FP Error |
|---|---|---|---|---|
| X-ent | 0.825 | 15.6 | 0.88 | 0.19 |
| X-ent BN | 0.835 | 15.9 | 0.84 | 0.14 |
| X-ent BN 32 | 0.836 | 15.5 | 0.83 | 0.12 |
| X-ent BN 8 | 0.829 | 16.0 | 0.85 | 0.15 |
| MDL | 0.830 | 16.4 | 0.81 | 0.12 |
| MDL BN | 0.808 | 16.9 | 0.90 | 0.25 |

Table 1 shows the median of the metrics measured on the 1,197 test cases. The distributions of these metrics are shown in Figure 3. In terms of the median DSC the best performing model was X-Ent BN 32. However, X-Ent BN produced very similar performance and required half of the GPU memory. As a result, X-Ent BN was chosen for further analysis.

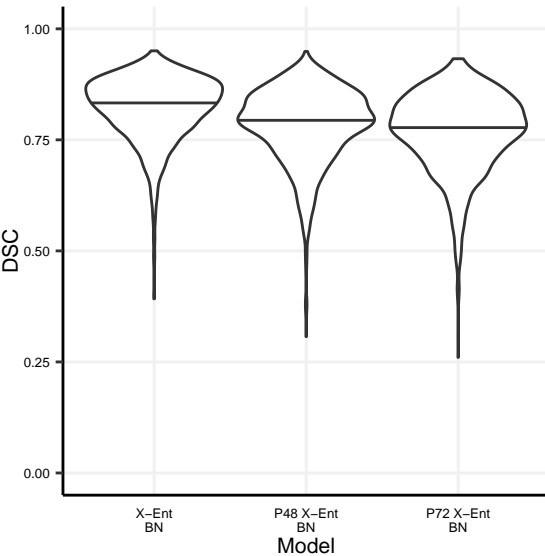

Figure 4: Distributions of the DSC for no padding (X-Ent BN) padding with segment size $48^3$ voxels (Patch 48 X-Ent BN) and padding with segment size $72^3$ voxels (Patch 72 X-Ent BN).

The variation of performance depending on the use of padding is shown in Figure 4. The median value of DSC for X-Ent BN, Patch 48 X-Ent BN and Patch 72 X-Ent BN was 0.835, 0.795 and 0.78 respectively.

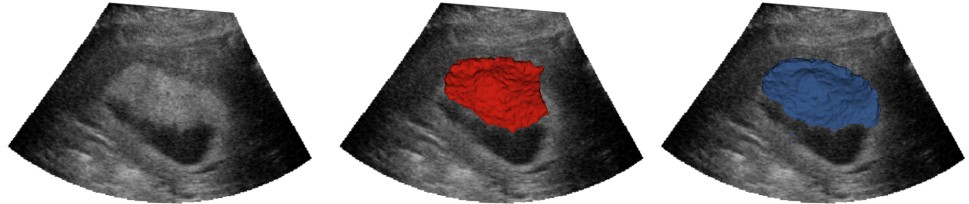

Figure 5: Sample segmentation of the placenta with a slice through the ultrasound volume shown. The red volume is the ground truth and the blue is the prediction of X-Net BN. The DSC was 0.836.

A sample segmentation is shown in Figure 5. The variation of the DSC with the overall quality score is shown for X-Ent BN in Figure 6. Pearson's correlation coefficient (r) of the DSC against the overall quality score is -0.28 (p < 0.05).

## 4 Conclusion

Six variations of a previously reported CNN were trained end to end. The best model with the highest median DSC during the training was chosen for each architecture and the performance on 1,197 cases was compared. There are many possible variations in hyper-parameters and doing an exhaustive search is not possible due to the prohibitive time training all potential options.

After the first epoch the lowest median DSS value obtained was for X-Ent. This is expected since batch normalisation is known to speed up training and was used in all models except for MDL. The DSC values of MDL BN were more volatile than the other models. MDL BN had a drop in median DSC from $0.75$ to $0.52$ between epochs three and four and another drop in median DSC between epochs six and seven.

The worst performing model was MDL BN with a DSC of 0.808 and the highest median Haudorff distance. The rest of the models had similar DSC values (0.825 to 0.836). Of these models there was some trade off in the target overlap and FP error where the MDL had the lowest of the target

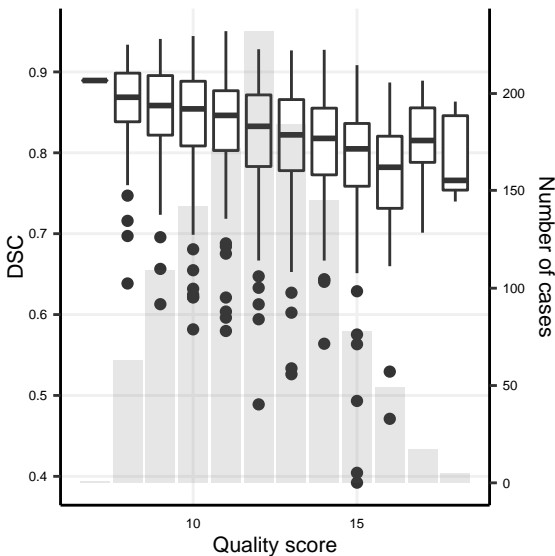

Figure 6: Box plot of the DSC for each overall image quality score. The number of cases in each box plot is shown as a column chart in the background.

overlap but also the lowest false positive error. The modified Dice loss layer was previously shown to give improved model performance [Milletari et al., 2016]. This observation was not corroborated by the results of this work. One possible explanation for this is that other hyper-parameters kept constant throughout are more suitable for cross-entropy loss than for modified Dice loss. The choice of learning rate and optimiser are two such possible hyper-parameters. Work by Sudre et al. [2017] suggested that the effect of the loss function was most important in cases where there is significant class imbalance. The class imbalance in segmenting the placenta in 3D ultrasound depends on the windowing by the operator when acquiring the image. The average foreground background ratio in this study was 17% which is much higher than those considered by Sudre et al. [2017] where background foreground ratios as low as 0.02% were considered.

Padding reduces the computational cost at the expense of homogenous confidence in voxel classification. In the two models where padded was used DSC was reduced. It is surprising that of the two models with padding the model with larger segment size and hence increased confidence of voxel classification, P72 X-Ent BN, had a reduced DSC when compared to P48 X-Ent BN. We believe this is due to the volatility of the training process. For comparison the best performing models from 10 epochs were chosen. This could possibly be criticised as an unfair comparison since the padded models will have much less input during an epoch because there is no overlap in the segments. The performance of the padded models could be improved by using reduced strides over the volume increasing the confidence of classification of voxels near the edge of a segment used. However, this would increase the computational cost which is the main benefit in using padding.

The variation of DSC values for X-Ent BN showed a correlation with the overall image quality score. Low quality images (ie with a high numerical score) tended to have a lower DSC and visa versa. Low quality ultrasound images make segmentation of the placenta difficult. Hence, the ground-truth segmentations on these images are less reliable than those on better quality cases. The decrease in DSC value on the lower quality cases will be due in some cases to errors in the ground-truth rather than a failure of the CNN. Obtaining accurate ground-truth particularly at this gestational age relies upon the clinical expertise of the operator and the quality of images that can be obtained. The images used in this study were collected several years ago using a machine that has since been superseded by two generations of ultrasound systems.

The goal of this work was to improve upon previously published methods for automatic segmentation of the placenta. A small increase in the quality of the segmentation was found. The results obtained using the modified Dice loss are not in agreement with those previously published. However, there are many confounding variables that could explain this apparent discrepancy. The variation of the DSC

with image quality suggests that improvements in image acquisition will allow for more accurate segmentation of the placenta. The tuning of the feature maps will enable faster analysis of 3D ultrasound volumes. This will allow for fast real-time analysis of placental volume that could enable screening of at risk pregnancies at an early gestational age.

**Acknowledgments**

The authors thank Prof. J. Alison Noble for her valuable input into the original placental imaging analysis that lead to the development of this work. We gratefully acknowledge the support of NVIDIA Corporation with the donation of the Tesla GTX Titan X GPU used for this research. PL, SC and research reported in this publication was supported by the Eunice Kennedy Shriver National Institute of Child Health and Human Development (NICHD) Human Placenta Project of the National Institutes of Health under award number UO1-HD087209. The content is solely the responsibility of the authors and does not necessarily represent the official views of the National Institutes of Health. GS is supported by a philanthropic grant from the Leslie Stevens' Fund, Sydney.

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
