# OpenReview forum: "Optimisation of a convolutional neural network to segment the first trimester placenta from 3D ultrasound scans"
_MIDL.amsterdam/2018/Conference — Submitted to MIDL 2018_

### Review · AnonReviewer2 · 2018-05-09
**A good paper but the argument about the batch-norm is shaky**

**Rating:** 3
**Confidence:** 3

**Review:**

Overall:
The paper considers a problem of 3D ultrasound scans for the pregnancy risk assessment. The authors propose to use a version of U-Net and improve on that using different loss function than the cross-entropy and different architectures (the batch normalization, padding, different sizes of feature maps). The paper is easy to follow and considers an important practical problem. Nevertheless, the proposed approach is very incremental to previous papers and, thus, the novelty is limited.

Strengths:
+ The paper is very well-written and easy to follow.
+ The considered problem is important and significant from the clinical point of view.
+ The proposed architecture is reasonable (except the batch normalization, please see remarks below).
+ The experiments are performed in detail and all results are properly analyzed.

Remarks:
* Major
- My main concern is the application of the batch normalization. The batch normalization is very useful if the considered problem does not contain even a slight domain shift. To be more precise, if a model is trained on data from one hospital and then it is tested on data from different hospital where a machine can introduce some artifacts or a measuring process is different, the batch normalization could potentially introduce a huge bias. I can even notice that, for instance, in Figure 2 where MLD BN and X-Ent BN 8 have higher variance of the validation performance than MLD and X-Ent 8, respectively.
- The authors claim that X-Ent BN 32 performs the best, however, the results given in Figure 3 do not provide any evidence on significant differences among the methods. In my opinion it is hard to say whether the batch normalization is helpful at all.
- The paper is very incremental comparing to previous papers and the proposed improvement using the batch normalization is not convincing. That is why, I decreased my decision by 1 point.

* Minor
- I miss a wall-clock time analysis during predicting. From the practical point of view it is important to know whether a model works in seconds, minutes or hours. In my opinion, showing these numbers would strengthen the paper and show that it is beneficial to apply deep learning (or AI in general) in medical imaging.

**Special Issue:**

No

---

### Review · AnonReviewer1 · 2018-05-09
**A well presented paper but detailing parameter tuning experiments which do not constitute novel work.**

**Rating:** 1
**Confidence:** 2

**Review:**

This paper describes experiments using deep learning for automatic segmentation of the placenta during pregnancy from 3D ultrasound (US).  The paper is in general well written and the clinical motivation for the method is established.  It would be interesting to determine (or report, if known) interobserver variability between experts in this task since it appears difficult to determine precise boundaries.  This could provide a useful benchmark for algorithm development.

My concern with this paper is that it is not sufficiently novel.  The description of the baseline network and its application to placental US is described in a separate submission to the MIDL conference.  The current work is concerned solely with gaining improvements on the reported Dice coefficients by making variations in the network architecture (loss function, number of feature maps, batch normalisation, padding).  These are the types of experiments that I would have expected the authors to carry out on their validation set as part of the original work.  They do not, in my opinion, have sufficient interest to merit a separate work.
Results from different architectures are not reported as being significantly different, nor do they appear to be so from the figures provided.  It appears that any improvements observed are likely due to natural variability in the data and there is no suggestion that the findings would generalise to other ultrasound datasets or different tasks.
The authors go to some effort to show that the Dice score is related to the image quality, which is interesting, but not unexpected.  This could be a minor addition to the other submission if it is not already contained there.

**Special Issue:**

No

---

> ### Comment · ~Padraig_T_Looney1 · 2018-05-14
> **Response**
>
> The separate abstract is based on a paper accepted for publication in the Journal of Clinical investigation insights. We are not permitted to publish over 400 words. Hence, we cannot merge this work with the other abstract.

---

### Review · AnonReviewer3 · 2018-05-10
**U-Net applied to the segmentation of 3D ultrasound. No clear contribution.**

**Rating:** 1
**Confidence:** 3

**Review:**

In this work, the authors evaluate the use of the U-Net method on the segmentation of the placenta from 3D ultrasound images. Several parameters of the U-net architecture and its optimizer are varied and comparative results are shown. Also, the performance of the best model is qualitatively assessed by an expert reader. The method is not compared against a published method. [There is an existing method cited in the paper,  but it's cited as "Anonymous [2018]".]
The authors discuss several choices in the implementation of the U-Net method (padding method, choice of the loss function, and the optimizer). This is interesting for the community. However, it could have been much more extensively done and I believe the authors should have picked a more state-of-the-art architecture. The quantitative results also don’t look very promising (DICE values of 0.80-0.83). I believe that a more state-of-the-art technique could have achieved better results and would have resulted in a more interesting read for the community.


**Special Issue:**

No

---

> ### Comment · ~Padraig_T_Looney1 · 2018-05-14
> **Response**
>
> Thank you for your comments. I believe that our DICE results are the state of the art for segmentation of the Placenta. Could you please give an example of a more state-of-the-art architecture for 3D segmentation?

---

### Decision · Program_Chairs · 2018-05-15
**Paper25 Acceptance Decision**

Reject